# The Impact of Pause-Internal Phonetic Particles on Recall in Synthesized Lectures

*Mikey Elmers[1], Éva Székely[2]*

[1]Saarland University, Germany
[2]KTH Royal Institute of Technology, Sweden
elmers@lst.uni-saarland.de, szekely@kth.se

## Abstract

We studied the effect of pause-internal phonetic particles (PINTs) on recall for native and non-native listeners of English in a listening experiment with synthesized material that simulated a university lecture. Using a neural speech synthesizer trained on recorded lectures with PINTs annotations, we generated three distinct conditions: a base version, a "silence" version where non-silence PINTs were replaced with silence, and a "nopints" version where all PINTs, including silences, were removed. Half of the participants were informed they were listening to computer-generated audio, while the other half were told the audio was recorded with a poor-quality microphone. We found that neither the condition nor the participants' native language significantly affected their overall score, and the presence of PINTs before critical information had a negative effect on recall. This study highlights the importance of considering PINTs for educational purposes in speech synthesis systems.

**Index Terms:** speech synthesis, pause particles, recall

## 1. Introduction

Pause-internal phonetic particles (PINTs) refer to various phenomena, such as breath noises (i.e., inhalations and exhalations), filler particles (FPs) like "uh" or "um", tongue clicks, and acoustic-phonetic silence. Studies have shown that PINTs can benefit listeners in both natural and synthetic speech. In natural speech, FPs have been found to improve the recall of story plot points [1] and the following word [2], while silent pauses have been found to improve the recall of the following word [3]. In synthetic speech, FPs have been shown to reduce the cognitive load for the listener [4], silent pauses have improved recall for digits [5], and breath noises have improved recall for sentences [6]. These findings collectively illustrate that PINTs can improve the recall for small contexts, such as words or sentences, in laboratory settings.

Research in speech synthesis for education is an important area of study that can yield benefits for both native speakers (NSs) and non-native speakers (NNSs). [7] found that NNSs comprehended synthetic sentences more easily than synthetic words, and that ratings were dependent on the listener's comprehension level. Additionally, [7] highlighted the need for investigating longer material lengths for speech synthesis. For example, when evaluating recall for larger contexts with real-world data, [8] found no recall effect for PINTs for both native and non-native listeners. Specifically, [8] used segments from English-language universities and found that PINTs reduced the recall for upcoming information. Therefore, we were interested in investigating how PINTs might influence synthesized speech in the same lecture setting, and whether their impact on recall was similar for both NSs and NNSs. Our present study repli-cated [8], using the same textual material but with a novel approach: the material was synthesized instead of being natural speech, and two instruction sets were created. One instruction set informed participants they were listening to synthesized speech, while the other instruction set told participants that the audio was recorded with a poor-quality microphone. Our investigation focused on the impact of PINTs on the recall of sections of synthesized lectures.

## 2. Method

We sourced lecture material from Open Yale Courses [9], a collection of free and open access courses provided by Yale University. The speaker used for this study had a significant occurrence of PINTs during their lectures, accounting for 40% of their total lecture time. We annotated a subset of these lectures to train the speech synthesis model. Fig. 1 shows a comparison of the same segment of speech from both the original and synthesized versions.

### 2.1. Stimuli

To generate the stimuli for the experiment, a neural text-to-speech (TTS) voice was created, using a method similar to the one described in [10]. This TTS system was trained using a PyTorch implementation[1] of the sequence-to-sequence neural TTS engine Tacotron 2 [11]. We used phoneme-level input for training and synthesis, which was obtained from the transcripts using the g2p_en package [12]. The training corpus was divided into segments, delineated by an inhalation breath on each end, where multiple breath groups were joined into utterances of at most 11 seconds. For the phonetic input, specific tokens for the different PINTs were added to allow exact prompted reproduction. The voice was trained for 70k iterations on top of the published read speech model, using 3 GPUs and a batch size of 28. The speech signal was decoded from the model output, using the neural vocoder HiFi-GAN [13]. The published model was finetuned for 1.33M iterations on the corpus.

The perceptual experiment material was generated one breath group at a time following [14, 15], with each segment starting and ending with an inhalation. Since the synthesizer is non-deterministic, we synthesized multiple versions and chose the version that included fewest distortions or errors[2]. Specifically, we avoided versions that included background noises, which occasionally occurred due to the varying audio quality of the speech corpus. After synthesizing all breath groups for

---

[1]https://github.com/NVIDIA/tacotron2
[2]Sample audio used for TTS comparison and perceptual experiment can be found at https://mikeyelmers.github.io/paper_ssw23ttsdemo/

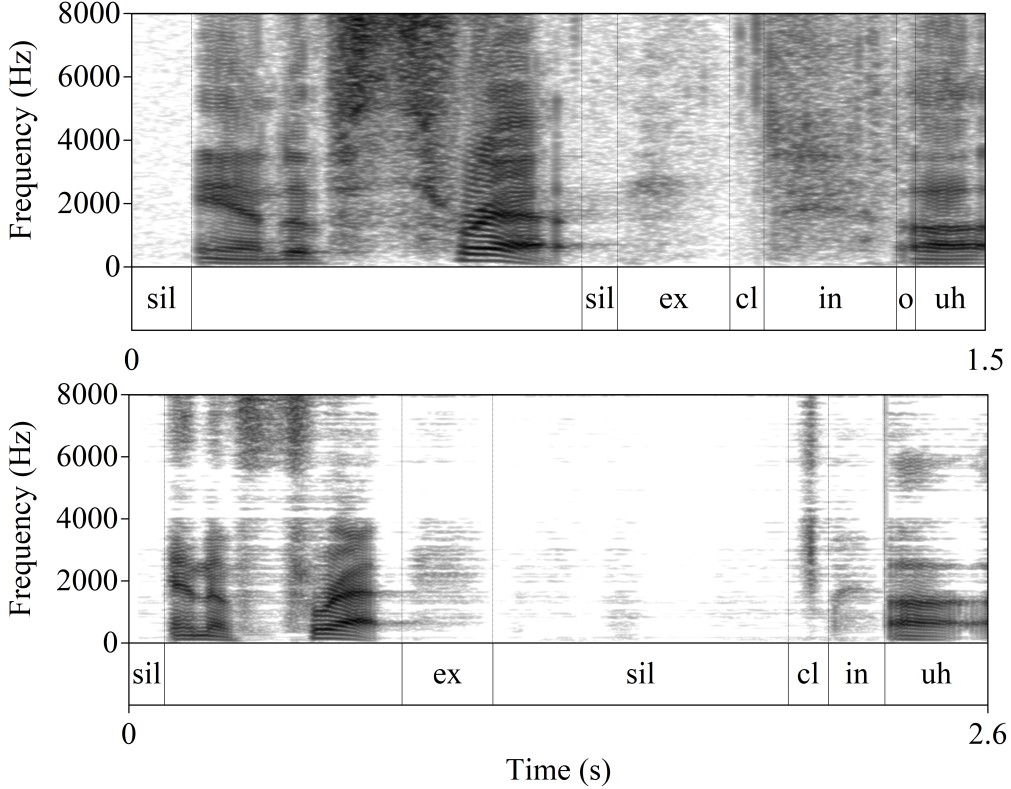

Figure 1: *Spectrogram comparison of original (top) and synthetic speech (bottom) for the same textual material. PINTs annotations: silence (sil), inhalation noise (in), exhalation noise (ex), filler particles (uh) and (um), tongue click (cl), and other (o).*

each lecture segment, we concatenated the segments together using Praat [16].

The study we replicated included three different conditions for the audio stimuli. We created the same conditions using a Praat [16] script, which either removed or replaced the PINTs material. For the silence condition, the non-silence PINTs were replaced with silence taken from the audio that was adjusted to match the duration of the removed material. As a result, the silence condition maintains the same duration as the base condition. The "nopints" condition removed all acoustic pauses. Both the base and silence conditions provide the same processing time, whereas the "nopints" condition allows for less processing time (see Fig. 2).

## 2.2. Perceptual Study

### 2.2.1. Participants

This study used a web-based experiment, designed with the Labvanced [17] platform, to present the audio stimuli and collect responses and questionnaire data from participants. Recruitment was carried out with the crowd-sourcing platform Prolific [18]. A total of 180 participants were recruited, including 90 monolingual English participants (mean age 40 years, age range 20–75 years) and 90 L1 German participants (mean age 33 years, age range 18–70 years), who received compensation for their participation. Three monolingual English participants and two L1 German participants self-reported hearing impairment and were excluded from the results.

### 2.2.2. Procedure

Participants listened to four sections of synthesized material, each approximately three minutes in length, and then answered two multiple-choice questions based on the content. The experimental material used the same textual content from [8], which extracted sections from full-length lectures. The stimuli were carefully chosen to ensure that all information needed to answer the question was present in the audio, i.e., no prior knowledge was required to answer the questions. For each audio segment,

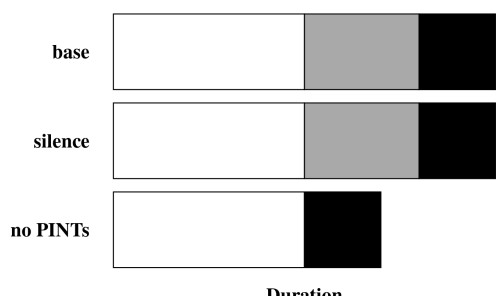

Figure 2: *Duration schematic for the three conditions, with speech material in white, PINTs in grey, and speech material that includes the answer depicted in black.*

one question was preceded by PINTs material, while the other was not. In the "nopints" condition, neither question contained PINTs material. An example question was, "According to Paul Fussell, what is the essential trope or rhetorical figure of World War One poetry?" Possible answers included: a) hyperbole, b) metaphor, c) oxymoron, d) irony. Participants only needed to understand the contents, as presented by the speaker, and did not require any encyclopedic or background knowledge to answer the questions. The study was balanced to ensure that the material preceding the first and second question was equally distributed, with or without PINTs.

Before beginning the experiment, participants received a set of instructions. They were informed that the study involved listening to four audio clips, each lasting approximately three minutes, and answering content-based questions immediately following each clip. The participants were instructed to use headphones and test their audio volume prior to commencing the experiment. There were told to not take any notes while listening to the audio clips. Half of the participants were informed that the audio was computer-generated audio, while the other half were informed that the recordings were made using a sub-optimal microphone, and contained some background noise. The latter group of participants were not informed that the audio was computer-generated. Participants were told that they would receive a score at the end of the experiment to incentivize them to perform well.

Following the listening task, participants were required to complete a questionnaire. The questionnaire included various demographic and language-related questions such as age, hearing impairment, L1, highest level of completed education, interest in the audio contents (1: very uninterested to 5: very interested), ease of following/understanding the speaker (1: very difficult to 5: very easy), and perception of the speaker's preparedness (1: very unprepared to 5: very prepared). The L1 German listeners were asked to provided their self-reported English proficiency and, if available, a test score. The total time for completing the listening task and questionnaire averaged between 15 to 20 minutes.

Each participant listened to one of three conditions. Specifically, one-third of the participants listened to four unmodified (i.e., "base" condition) audio clips, another one-third listened to four audio clips from the "silence" condition, and the remaining one-third listened to four clips from the "nopints" condition. The textual material was the same across all conditions, but the order of the four audio clips was randomized to prevent potential order effects. Participants saw "Listen closely!" displayed on their screen while the audio played, and each audio clip was played only once.

# 3. Results

R [19] (Version 4.1.2) was used to perform both descriptive and inferential statistics[3]. Prior to analysis, the data was preprocessed using the dplyr [20] (Version 1.1.1), stringr [21] (Version 1.5.0), and tidyr [22] (Version 1.3.0) packages. The homogeneity of variance assumption was assessed using Levene's test from the car [23] (Version 3.1.2) package. Cohen's d was calculated using the effsize [24] (Version 0.8.1) package. Posthoc comparisons after conducting the Kruskal-Wallis rank sum test were performed using Dunn's Test from the dunn.test [25] (Version 1.3.5) package. The statistical models were analyzed

---

[3]All data and code for the results can be accessed at `https://github.com/MikeyElmers/paper_ssw23`

using either linear regression or with binomial generalized linear mixed models (binomial GLMMs) implemented through the lme4 [26] (Version 1.1.32) and lmerTest [27] (Version 3.1.3) packages. The Akaike Information Criterion (AIC) [28] was used to compare models and choose the best fit model with the lowest AIC. Visualizations were generated using ggplot2 [29] (Version 3.4.2).

## 3.1. Duration and Count

We compared the count and duration information of the four synthesized passages to the original versions. The originals had a longer PINTs duration, a longer total duration, and a higher PINTs proportion (see Table 1). However, while the synthesized passages were shorter overall, the PINTs proportions were comparable. Both versions were similar to the speaker's overall PINTs profile proportion of 40%. Table 2 contains count information for the individual PINTs grouped by condition. The original material had more silences, exhalations, and "other" particles. Both versions had a similar number of inhalations, filler particles "uh" and "um", and tongue clicks.

Table 1: *Duration information for the original and synthesized material. Values include all four passages. Total PINTs duration (PINTs dur) and total audio duration (total dur) are reported in seconds. Proportion (prop) is expressed as a percentage value out of 100%.*

| condition | PINTs dur | total dur | prop |
|---|---|---|---|
| *original* | 314.45 | 761.48 | 41.29 |
| *synthesized* | 273.68 | 705.23 | 38.81 |

Table 2: *Count information for the original and synthesized material. The values include all four passages. The following labels were investigated: silence (sil), inhalation (in), exhalation (ex), filler particles (uh) and (um), tongue click (cl), other (o).*

| condition | sil | in | ex | uh | um | cl | o |
|---|---|---|---|---|---|---|---|
| *original* | 656 | 234 | 65 | 210 | 30 | 43 | 82 |
| *synthesized* | 288 | 232 | 9 | 212 | 30 | 45 | 2 |

## 3.2. L1 Comparison

Participant performance was determined by the number of correctly answered questions, with a potential maximum score of 8 (1 point for each correct answer). We compared total score means grouped by L1. The normality assumption for the dependent variable was violated, as determined by the Shapiro-Wilk Normality test ($p < 0.05$). The assumption of equal variances was met, as verified by Levene's test for homogeneity ($p > 0.05$). Since the sample sizes exceeded 30, a parametric test was used following the central limit theorem, despite the violation of normality. An independent samples t-test compared the mean scores of the monolingual English group (M = 5.31, SD = 1.67) and the L1 German group (M = 5.57, SD = 1.64). The findings revealed *no* significant difference between the two groups (t(173) = -1.03, $p > 0.05$, d = -0.16), indicating that the L1 German group and monolingual English group performed similarly.

### 3.3. Condition and Group Comparison

Given that L1 was not a significant factor, we proceeded to investigate the effect of condition. Total scores grouped by condition are summarized in Table 3. Participants were informed that they were either listening to computer-generated audio (in half of the conditions) or that the speaker was using a poor-quality microphone (in the remaining half). Notably, the highest mean score was obtained in the silence condition where participants were told that the audio was computer-generated. The lowest mean score was observed in the silence condition where participants were told that the speaker used a poor-quality microphone. This pattern was not universal, but in general, when participants were told the audio was computer-generated, the total score was higher than when they were told the audio came from a poor-quality microphone, except for the "nopints" condition.

Table 3: *Descriptive statistics for total score (possible maximum score of 8) for the different conditions. Participants were told that the audio was computer-generated in the conditions with the subscript "cg".*

| condition | mean | median | sd | N |
|---|---|---|---|---|
| $silence_{cg}$ | 5.77 | 5 | 1.41 | 31 |
| $nopints$ | 5.71 | 6 | 1.61 | 28 |
| $base_{cg}$ | 5.61 | 6 | 1.64 | 28 |
| $base$ | 5.45 | 5 | 1.78 | 29 |
| $nopints_{cg}$ | 5.34 | 5 | 1.72 | 29 |
| $silence$ | 4.77 | 5 | 1.68 | 30 |

We conducted an analysis of variance (ANOVA) to investigate the mean differences in total score between conditions, with condition as a fixed factor. The normality assumption was violated in the following comparisons as indicated by the Shapiro-Wilk test, while Levene's Test did not reveal any significant differences in variances across groups. Given that the sample sizes exceeded 30, satisfying the central limit theorem, we utilized parametric tests. The results did not indicate any significant effect of condition on total score ($F(5, 169) = 1.50$, $p > 0.05$). Post-hoc comparisons using pairwise t-tests with Bonferroni correction did not reveal any significant differences between individual groups. However, the Bonferroni correction is conservative by nature. A two-sample t-test was conducted to compare the mean total score between the $silence_{cg}$ (M = 5.77, SD = 1.41) and the silence (M = 4.77, SD = 1.68) conditions. The results showed that the $silence_{cg}$ condition had a significantly higher total score ($t(59) = 2.55$, $p < 0.05$, d = 0.65). We conducted additional analyses by grouping conditions based on audio type (i.e., base, silence, and nopints) and whether participants were informed that the audio was computer-generated (i.e., cg group and non-cg group). However, these analyses did not reveal any significant differences in mean total score when grouping by audio type ($F(2, 172) = 0.44$, $p > 0.05$) or by computer-generated instruction type ($t(173) = 1.12$, $p > 0.05$, d = 0.17).

### 3.4. Preceding Material

We explored the impact on recall based on whether the question material was preceded by PINTs. Table 4 demonstrates that answers immediately preceded by PINTs material had an overall lower score. The nopints and $nopints_{cg}$ conditions, which did not contain any PINTs information, were excluded from this analysis. Scores were out of 1, rather than 8, since the evaluation was done on a by-question basis rather than the subject's collective score. A score of 1 indicated a correct response, and 0 an incorrect response. Due to violations of both normality, as indicated by the Shapiro-Wilk test, and homogeneity of variances, as indicated by Levene's test, we used the non-parametric Wilcoxon rank sum test. Our analysis revealed a significant difference between the conditions where PINTs material immediately preceded the question, and those where it did not (W = 132396, $p < 0.001$). Specifically, the mean score for questions preceded by PINTs material was significantly lower than those without any preceding PINTs information. These results suggest that the presence of PINTs information immediately before key information, may have a detrimental effect on recall performance.

Table 4: *Descriptive statistics, including mean and standard deviation, based on whether the question material was immediately preceded by PINTs material.*

| precede | mean | sd |
|---|---|---|
| $no$ | 0.77 | 0.42 |
| $yes$ | 0.58 | 0.49 |

We utilized a binomial generalized linear mixed effects model to investigate the relationship between score (0 or 1) and preceding PINTs, L1, condition, and questionnaire variables. The model with the best fit, as determined by the lowest AIC, was $glmer(score \sim precede + interest + (1 \mid id), family = binomial)$. This model predicted score based on whether the answer was preceded by PINTs information and interest level as fixed effects, with the subject as a random effect. This model outperformed alternative models that incorporated L1 status, condition, or other questionnaire variables. Our findings indicated that both preceding PINTs and interest level significantly predicted score. The intercept was significant ($\beta = 0.84$, $p < 0.001$), and whether the answer was preceded by PINTs material had a significant negative effect ($\beta = -0.96$, $p < 0.001$). Interest level had a significant effect on score, with the highest levels of interest being the strongest predictors ($\beta = 1.80$, $p < 0.001$ for interest level 5; $\beta = 0.98$, $p < 0.001$ for interest level 4; $\beta = 0.42$, $p > 0.05$ for interest level 3; $\beta = 0.38$, $p > 0.05$ for interest level 2). These results suggest that only the highest levels of interest (4 and 5) are associated with higher odds of a positive outcome on score. In summary, our findings indicate that when holding all other variables constant, the odds of a positive outcome on score decrease when PINTs precede, while the highest levels of interest are associated with higher odds of a positive outcome on score.

### 3.5. Interest

Participants rated their level of interest in the audio contents on a scale of 1 (very uninterested) to 5 (very interested), with a mean rating of 2.73. An independent samples t-test was conducted to compare the mean interest ratings between the monolingual English group (M = 2.70, SD = 1.22) and the L1 German group (M = 2.76, SD = 1.13). The results indicated no significant difference between the two groups ($t(173) = -0.33$, $p > 0.05$, d = -0.05).

Interest was further examined by condition (see Table 5). We used an ANOVA to investigate mean differences in interest between the different conditions. The results revealed a signifi-

cant effect of condition on interest (F(5, 169) = 2.45, $p < 0.05$). Post-hoc pair-wise t-tests with Bonferroni correction showed a significant difference between the nopints condition and the base condition ($p < 0.05$). Both the L1 and condition comparisons violated normality, as indicated by the Shapiro-Wilk test. However, homogeneity of variances was maintained, as indicated by Levene's test. As a result, we opted for parametric tests since both comparisons satisfied the central limit theorem.

Table 5: *Mean interest values for the different conditions. Participants were told that the audio was computer-generated in the conditions with the subscript "cg".*

| condition | mean | sd |
|---|---|---|
| $nopints$ | 3.25 | 1.17 |
| $sil$ | 2.90 | 1.03 |
| $sil_{cg}$ | 2.77 | 1.09 |
| $base_{cg}$ | 2.64 | 1.16 |
| $nopints_{cg}$ | 2.59 | 1.18 |
| $base$ | 2.24 | 1.27 |

We used linear regression models to investigate the relationship between total score and L1, condition, and the questionnaire variables. The model with the lowest AIC included interest and instruction type (whether participant were told the audio was computer-generated) as fixed effects: $lm(total\ score \sim interest + cg)$. This model was statistically significant (F(5, 169) = 5.76, $p < 0.001$) and explained 14.56% of the variance in total score ($R^2 = 0.1456$). Our results indicated that interest level had a significant effect on total score, with the highest level of interest being the strongest predictor ($\beta = 2.42$, $p < 0.001$ for interest level 5; $\beta = 1.05$, $p < 0.01$ for interest level 4; $\beta = 0.42$, $p > 0.05$ for interest level 3; $\beta = 0.13$, $p > 0.05$ for interest level 2). However, instruction type did not have a significant effect on total score ($\beta = 0.38$, $p > 0.05$). The intercept was significant ($\beta = 4.72$, $p < 0.001$). The adjusted R-squared of the model was 0.12, with a residual standard error of 1.55. These findings suggest that the level of interest in the audio content has a significant effect on total score, but instruction type does not have a significant effect on total score.

### 3.6. Correlation

Pearson correlations for participants' total score and their questionnaire responses are presented in Table 6. The questionnaire assessed how easy it was to comprehend the speaker, level of interest in the lecture content, and evaluated the speaker's preparedness. Ratings were made on a 5-point Likert scale. Age, ease, and preparedness were weakly correlated with total score, while interest was moderately correlated (t(173) = 4.56, $p < 0.001$, r = .33). The correlation between total score and interest was stronger for monolingual English participants than for L1 German participants. Overall, the higher the participant's interest level, the better their total score.

## 4. Discussion

Compared to the original version, the synthesized version struggled to generate short silences, often found adjacent to non-silence PINTs (i.e., edge silences), resulting in fewer silences. Similarly, the counts for exhalations and the "other" category also decreased in the synthesized version, possibly due to the scarcity of exhalations in the data. The "other" category was

Table 6: *Pearson correlation coefficients for participants' total score and their questionnaire responses. Includes correlation information for all participants (Both), L1 German (DE), and monolingual English (EN).*

| participants | age | ease | interest | prep |
|---|---|---|---|---|
| *Both* | 0.20 | 0.15 | 0.33 | 0.10 |
| *DE* | 0.15 | 0.07 | 0.25 | −0.00 |
| *EN* | 0.30 | 0.24 | 0.39 | 0.19 |

not included during the synthesizer's training due to the diversity of phenomena within the category. Nevertheless, the synthesizer generated some "other" labels without any explicit inclusion. Despite these differences, the synthesizer maintained similar counts for inhalations, filler particles "uh" and "um", and tongue clicks. However, the synthesizer occasionally produced multiple PINTs from a single label, leading to a higher count of "uh" and tongue clicks than was present in the original material. Overall, it was unexpected to find that the synthesized version closely modeled some of the counts, given that the output is not deterministic.

Our findings indicated that the participants' total score was not affected by their language background, whether they were monolingual English or L1 German, when listening to English-language content. This is a favorable outcome that suggests synthetic speech could be an effective equalizer for educational purposes, as both NSs and NNSs performed similarly. Additionally, we also observed significant differences in total score between the condition where participants were told the audio was synthesized ($sil_{cg}$), and the condition where participants were told the audio came from a poor-quality microphone (sil). This discrepancy may be due to participants being more lenient when they knew that the material was synthesized, as opposed to those who might be more critical assuming it was from a human speaker.

Our study found similar results to the study we replicated [8], that PINTs preceding key information lowered recall. This highlights the need for future research to determine when PINTs can be beneficial for recall. However, neither study was able to replicate the benefits of PINTs observed in single sentence laboratory settings. It is possible that the PINTs profile of the speaker we used to train the TTS is an outlier. Lecture recordings from a speaker who uses PINTs to a lesser extent may reveal recall benefits. One limitation of using a single speaker is the difficulty in comparing how listeners perceive different realizations of the same PINTs. Moreover, it is challenging to isolate individual PINTs for analysis in spontaneous speech recordings, where many PINTs co-occur. In this experiment, we treated all PINTs equally, despite each PINT having different realizations that may impact recall differently.

Participants who rated their interest in the audio content as high (4 or 5) had a significantly higher total score, and this was reflected in a moderate correlation between interest and total score. Contrary to expectations, the instruction type did not have a significant effect on the total score, suggesting that whether the participants knew the material was synthesized or not, did not impact their performance. When comparing mean interest scores by condition, it was unexpected to find the "nopints" condition had a higher interest score than the "base" condition, despite the audio artefacts resulting from the removal of the pause material. One possible explanation is that, again, the speaker used for training the TTS model used too

many PINTs during their lectures, which might have resulted in a less engaging experience.

For most levels of proficiency, [30] found that pauses helped comprehension more than speaking at a normal rate or artificially slowing the speaking rate. However, there is a threshold, beyond which pauses have a negative effect on comprehension [30, 31]. It is possible that the PINTs used in this study exceeded the threshold and became a detriment to recall. These results indicate that the impact of PINTs in synthesized speech should be carefully considered when recall of information is important, such as in education, for both native and non-native listeners.

## 5. Conclusion

This work replicated the experimental methodology used in [8] by evaluating the effect of pause-internal phonetic particles on recall, but used synthesized speech rather than natural speech. Our neural speech synthesizer was trained on labeled PINTs annotation, and generated experimental stimuli. First, we compared the count and duration information for the natural and synthetic speech. We found that the PINTs proportion for the synthesized version was similar to the natural version. Next, we conducted a perceptual experiment with three conditions: 1) a base version, 2) a "silence" version where all non-silence PINTs were replaced with silence of the same duration, and 3) a "no PINTs" version where all PINTs material was removed. Similar to [8], recall of key information was lower when the material was preceded by PINTs. Importantly, participant's recall was not influenced by their first language. Half of participants were told they would hear computer-generated audio, while the other half was told that the audio was recorded with a poor-quality microphone. Overall, the instruction type did not have a significant effect on recall, indicating that participants didn't adjust their expectations based on whether or not they knew the audio was synthesized. Participant's also provided their subjective evaluations, such as interest level, via a questionnaire. Higher levels of interest resulted in a significant positive effect on the recall of key information.

This work showcased that PINTs are widely present in real-world lectures, and that the presence of PINTs lowered the recall of key information for synthetic speech. The lectures used as training data incorporated a high number of PINTs. An interesting area for future research is whether these conclusions hold with a lower, potentially less distracting number of PINTs. Especially since these findings contrast with the results of single-sentence laboratory experiments, indicating that additional work is required to tease apart the influence of PINTs on recall.

## 6. Acknowledgments

This research was funded in part by the Deutsche Forschungsgemeinschaft (DFG, German Research Foundation) – Project-IDs MO 597/10-1 & TR 468/3-1, in part by the Swedish Research Council project Perception of speaker stance (VR-2020-02396) and by the Riksbankens Jubileumsfond project CAPTivating (P20-0298).

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
