# OpenReview forum: "The Impact of Pause-Internal Phonetic Particles on Recall in Synthesized Lectures"
_Interspeech.org/2023/Workshop/SSW — SSW12_

### Official Review · Reviewer_mTnE · 2023-05-29
**The Impact of Pause-Internal Phonetic Particles on Recall in Synthesized Lectures**

**Rating:** 4
**Confidence:** 3

**Review:**

The manuscript describes an experiment to determine whether - essentially - non-speech components have an effect on subsequent recall of information.  Information is conveyed to participants using synthetic speech, which allows the non-speech components to be inserted or not as required by the experiment.

Pros:
The experimental process in interesting.  That synthetic speech is now good enough to enable this kind of experiment open up a new field in use of synthesis in such contexts.  The results are analysed with particular statistical rigor.

Cons:
It is a little difficult to comment as my own background is in the synthesis technology itself rather than the use of synthesis.  Personally I found the results very difficult to discern, with the emphasis on significance tests clouding the understanding of the experiments themselves.
In looking for a summary of the findings, there is no conclusion.  Rather, the best summary is in abstract.  The overall conclusion is perhaps negative - that the presence of "PINTS" reduces the overall recall.  In fact it is laudable that the authors present a negative result, but it is not profound; I don't think it has strong implications for the speech synthesis community.

There is a question of suitability for the workshop.  For me, SSW is more about the synthesis technology rather than applications like this.  The paper may be better received at forum on psychology or social science.  A similar paper appears to have been submitted to Interspeech, which also feels more appropriate.  Then again, the authors perhaps mean to feed back the concept of this kind of experiment to the synthesis community, and I find that perfectly valid.

Clarity is fine, aside from the above comment about the significance tests clouding the results.  References seem good.

---

> ### Author Response · Authors · 2023-07-13
> **Paper Conclusion Modification**
>
> Thanks for the wonderful feedback. We've included a conclusion that better summarizes and clarifies the results.

---

### Official Review · Reviewer_5hmP · 2023-06-05
**Thorough evaluation of PINTS on recall**

**Rating:** 7
**Confidence:** 4

**Review:**

This is an interesting paper on the effect of pause-internal phonetic particles. The authors do a good job explaining all the statistical analyses performed, but it does make it a bit cumbersome to read. At first I was surprised by the findings that they do not help but rather hurt recall. But then I listened to the natural and synthesized example in the supplementary material, and found that the speaker giving the lecture had a very large number of PINTs which in my opinion is already making it harder to process. The synthesis did have some artefacts, but I don't think they would have affected the recall.

---

### Decision · Program_Chairs · 2023-06-14

**Decision:**

Accept

**Comment:**

SSW2003 received 45 papers. The acceptance rate is 82%. We are pleased to inform you that your paper has been accepted by the SSW2023 Program Committee. Please read the reviews carefully and submit your camera-ready paper by June 28th. Most of reviewers performed a detailed review. Please answer to their questions and take into account their comments.
Since your paper received a score below 5/9 that is strongly argued by the reviewers, note that the Program Committee will check if your manuscript has been significantly changed to specifically consider their remarks. Note that camera-ready papers are credited with one extra page to allow authors to consider reviewers’ suggestions. So max 7 pages in total including figures & refs.
The deadline for submitting the revised version (with full non anonymized authors and refs!) is 28th June.